

# The Dynamical Impact of Rossby Wave Breaking upon UK PM10 Concentration

C. P. Webber[1], H. F. Dacre[1], W. J. Collins[1], and G. Masato[1]

[1]Department of Meteorology, University of Reading, Berkshire, RG6 6AX

*Correspondence to:* C. P. Webber (c.p.webber@pgr.reading.ac.uk)

**Abstract.**

Coarse particulate matter (PM10) has long been understood to be hazardous to human health with mortality rates increasing as a result of raised ground level concentrations. We explore the influence of synoptic scale meteorology on observed PM10 concentration ([PM10]) using Rossby Wave Breaking (RWB). Meteorological re-analysis data for the winter months (DJF)

between January 1999 and December 2008 and observed PM10 data for three urban background UK (Midland) sites, were analysed. Three RWB diagnostics were used to identify RWB that had significant influence on UK Midland PM10. RWB events were classified according to whether the RWB was cyclonic or anticyclonic in its direction of breaking and whether the RWB event was influenced more by poleward or equatorial air masses.

We find that there is a strong link between RWB events and UK [PM10]. Significant increases (p<0.05) in UK [PM10] were

seen one day following RWB occurring in spatially constrained Northeast Atlantic/ European regions. Analysis into episodic PM10 exceedance events shows increased probability of [PM10] exceedance associated with all RWB subsets. The greatest probability of exceeding UK [PM10] threshold was associated with cyclonic RWB preceded by anticyclonic RWB forming an Omega block synoptic pattern. This mechanism suggests an easterly advection of European PM10 followed by prolonged stagnant conditions within the UK and led to an almost 3-fold increase in the probability of the UK Midlands exceeding a

hazardous [PM10] threshold.

## 1  Introduction

The influence of large scale flow patterns upon pollutant concentrations within the UK is often overlooked in relation to the contribution of local sources, sinks and smaller scale boundary layer transport processes. However some source receptor (Charron et al., 2013) and source attribution (Yin and Harrison, 2008) studies have shown the influence that external (continental

European) sources have upon pollutant levels within the UK. Both studies highlight the strong influence of large scale advected inorganic species on UK particulate matter (PM) with an aerodynamic diameter $\leq 10\,\mu\mathrm{m}$ (PM10). The aim of this study is to quantify the relationship between large scale atmospheric flow patterns and UK PM10 concentration ([PM10]) and to understand the mechanisms driving this relationship.

PM was chosen as the subject of analysis, due to its detrimental impacts upon human health (Kappos et al., 2004). The

health impacts of PM depend greatly on its size and composition, with finer particles able to travel further down the human





respiratory tract, subsequently increasing its anthropogenic toxicity. Throughout the period of this study, the monitoring of fine PM (PM2.5) was spatially poor and so PM10 was analysed.

Previous research has shown the dominant synoptic regimes influencing the UK, which can be generally described by zonal flow and blocked flow (Pope et al., 2015 and McGregor and Bamzelis, 1999). McGregor and Bamzelis, 1999 highlighted that UK [PM10] was significantly reduced by the presence of zonal flow entering the UK, while blocked regimes influenced an increase in UK [PM10]. Zonal and blocked flow patterns are largely determined by the absence or occurrence of Rossby wave breaking (RWB) within the Northeast Atlantic/ European region (Woollings et al., 2008).

RWB is the large scale overturning of air masses in the upper troposphere, caused by mechanical shear induced by the mid latitude eddy driven jet (EDJ). RWB leads to blocked flow within the Northeast Atlantic/ European region, through the generation of a climatological anomalous high pressure system over Western Europe. This high pressure anomaly can influence the UK through the generation of weak winds or easterlies over the region. We hypothesise therefore that the generation of anomalous easterlies into the UK will facilitate the advection of PM10 from the European continent and subsequently raise UK [PM10]. In addition to enhancing European PM advection into the UK, RWB is also shown to disrupt the mid-latitude EDJ and subsequent northeast Atlantic storm track. Associated with the storm track are loss mechanisms effective in removing ambient PM. Increased precipitation is the greatest PM sink associated with enhanced storm activity, while the introduction of a cleaner westerly maritime air mass acts to lower background UK [PM10] (McGregor, 1999). Furthermore, enhanced storm activity is associated with elevated surface wind velocity (horizontal and vertical), which enables the transport of surface PM10 away from the surface source region. This advection of PM10 from the source region results in a reduction of surface [PM10], as the pollutant becomes more disperse.

In this study RWB is diagnosed on the tropopause, which can be identified dynamically using the potential vorticity (PV) = 2 PVU surface (where 1 PVU = $1x10^{-6}$ $m^2\,s^{-1}\,K\,kg^{-1}$). The diagnostic used to diagnose RWB is theta on the 2-PVU surface ($\theta$-2PVU). $\theta$ is materially conserved following frictionless, adiabatic (isentropic) flow and to a good approximation, the extratropical atmosphere behaves as an adiabatic and frictionless fluid (Hoskins, 1997). The $\theta$-2PVU surface has been used on numerous occasions to identify extratropical blocking resultant from RWB (Pelly and Hoskins, 2003; Shutts, 1983; Tyrlis and Hoskins, 2008). The significance of the 2-PVU isertelic surface is that at the tropopause, the isentropic gradients are vertically concentrated and with one surface, upper level synoptic features can be illustrated (Morgan and Nielsen-Gammon, 1998).

The climatological mean sea level pressure (MSLP) response to RWB is a MSLP dipole formed of an anomalous high/low pressure lobe to the north/south of the centre of overturning (Masato et al., 2013 - hereafter M13). The pressure dipole results from the meridional advection of upper level air masses with anomalous potential vorticity (PV) characteristics. Equatorward air masses have lower PV due to a lower planetary vorticity component, while poleward air masses experience high planetary vorticity and subsequent high PV. When these air masses are meridionally overturned such as in RWB, anomalously high/low PV is advected south/north, resulting in cyclonic/anticyclonic tendencies respectively.

We hypothesise that RWB will impact UK [PM10] through the displacement of the EDJ and removal of the primary PM10 loss mechanisms. Following this, if RWB is to become influential upon raising [PM10] above a background concentration, an





element of persistence is required (Cattiaux et al., 2010). If a persistent atmospheric block is produced, this will facilitate the accumulation of either local or advected PM10. Furthermore, easterlies associated with RWB may advect continental PM into the UK and subsequently increase local [PM10].

## 2 Methodology

65  Within this study, data was analysed for the winter months (DJF) as it has been previously found that RWB is the primary mechanism for persistent high pressure over the UK region, during the winter (Pelly and Hoskins, 2003). The extent of all data used within this paper are January 1999 - February 2008 (DJF).

### 2.1 Meteorological large-scale flow diagnostics

M13 developed a 2D blocking index (BI) which measured the meridional extent to which the constant $\theta$-2PVU contour has overturned (Eq. 3). A positive BI value indicates overturning of the $\theta$-2PVU contour, representing blocked flow, while negative BI values represent zonal flow. Equations 1/2 represent mean $\theta$ ($\overline{\overline{\theta}}$) in the $15^o$ latitude ($\phi$) to the north/south (n/s) of the grid

point of interest ($\phi_0$) respectively for every grid point, with i and j representing longitudinal and latitudinal coordinates and $\triangle\phi$ is given in km.

$$\overline{\theta}^n_{i,j,t} = \frac{2}{\triangle\phi} \int\limits_{\phi_0}^{\phi_0+\triangle\phi/2} \theta_{i,j,t}d\phi \tag{1}$$

$$\overline{\theta}^s_{i,j,t} = \frac{2}{\triangle\phi} \int\limits_{\phi_0-\triangle\phi/2}^{\phi_0} \theta_{i,j,t}d\phi \tag{2}$$

$$BI_{i,j,t} = \overline{\theta}^n_{i,j,t} - \overline{\theta}^s_{i,j,t} \tag{3}$$

A second diagnostic, the Direction of Blocking Index (DB index), is used to determine the zonal gradient of the $\theta$-2PVU surface. The metric diagnoses whether an overturning, occurs in either a cyclonic (CRWB) or anticyclonic (ACRWB) direction.

Variations of this diagnostic have been used within multiple studies (Weijenborg et al., 2012, M13 and Pelly and Hoskins, 2003), but the 2D DB index described in M13, is used within this study. The DB index is calculated using Eq. 5 and is a measure of the horizontal gradient of $\theta$ centred on the grid point of interest. The equation describes $\overline{\overline{\theta}}$ to the east of the grid point of interest subtracted from $\overline{\overline{\theta}}$ to the west of the grid point. A positive DB Index value indicates that the overturning will rotate with anticyclonic motion, while for negative values, a cyclonic motion is diagnosed. Weijenborg et al., 2012 showed that

CRWB and ACRWB occur climatologically in different regions. It was shown by Weijenborg et al., 2012 that the meridional zonal shear imparted on the background flow by the EDJ generates cyclonic/anticyclonic motion to its north/south, resulting in CRWB/ACRWB occurring respectively.





The diagnostic detailed in Eq. 6 is the Relative Intensity of air masses Index (RI index), which compares the relative quantities of warm equatorial air to cold poleward air influencing a RWB event. This metric measures the $\overline{\theta}_{i,j\,t}$ anomaly at a given time (t) from the 44 year climatological ERA-Interim winter (DJF) value for that grid point ($\theta_{i,j}^*$). If this value is positive/ negative it indicates that the overturning event is majority warm/ cold influenced. This metric helps to determine the strength of the low/high PV lobes to the south/north of a RWB event. The strength of the PV lobes will directly influence the strength of the associated pressure anomalies within the associated MSLP dipole.

For both the DB and RI indices, any magnitude within the range of $\left(-0.2\,\mathrm{K\,km^{-1}} \le x \le 0.2\,\mathrm{K\,km^{-1}}\right)$ is determined unclassified as described in M13. All blocking metrics used within this study were generated using ECMWF ERA-Interim reanalysis data. The data has been temporally filtered as 24 hr mean values, generating daily resolved large-scale flow diagnostics.

$$\overline{\theta}_{i,j,t} = \frac{\overline{\theta}_{i,j,t}^n + \overline{\theta}_{i,j,t}^s}{2} \tag{4}$$

$$DB_{i,j} = \overline{\theta}_{i-1,j,t} - \overline{\theta}_{i+1,j,t} \tag{5}$$

$$RI_{i,j} = \overline{\theta}_{i,j,t} - \theta_{i,j}^* \tag{6}$$

## 2.2 PM10 data

PM10 data from three UK Midland PM10 monitoring sites: Birmingham Central, Leicester Central and Leamington Spa were used for winter months (DJF) between January 1999 and February 2008. The three sites used are all classified by DEFRA (DEFRA, 2014) as urban background sites, each representative of their urban environment. The primary benefit of using such a site is that the majority of the UK's population live within urban areas. To best represent the UK, a sampling region was selected within the centre of the UK, to account for influences from all directions. Due to the constant development of urban areas, each of the monitoring sites are influenced by nearby traffic (Harrison, 1997), roadworks, construction and other anthropogenic activities and subsequently spikes in the data may affect each site independently. Inconsistent, localised [PM10] spikes were identified for each site and removed when the daily mean [PM10] tendency ($[PM10]_t$-$[PM10]_{t-1}$) at two sites contrasted that of the third site. All tendency differences that lay within the top 10% of tendency differences from the entire dataset were removed as locally influenced spikes. The sites used, formed a tri-site super site and the daily [PM10] from each site was averaged to form one representative running mean. In a situation where a site has data removed due to irregularities in tri-site comparisons, PM10 from the remaining two sites were used. [PM10] datasets inherently have log-normal distributions, therefore all [PM10] values quoted within this study will be of the form $\log_e$[PM10] to ensure that the tri-site dataset has a normal Gaussian distribution.



A daily mean [PM10] ($\overline{\text{PM10}}$) exceedance has been defined in this study, when $\overline{\text{PM10}}$ exceeds the threshold of 29.72 $\mu\text{g m}^{-3}$ ($\log_e[\text{PM10}] = 3.39$). This threshold has been determined following epidemiological cohort studies undertaken as part of the European Study of Cohorts for Air Pollution Effects (ESCAPE) project. Studies within the ESCAPE and APHEA2 projects highlighted that detrimental anthropogenic impacts were shown to be significant for [PM10] that exceeded a sites $\overline{\text{PM10}}$ by 10 $\mu\text{gm}^{-3}$ (Gehring et al., 2013; Katsouyanni et al., 2001). For PM2.5 levels the same conclusions were reached (Beelen et al., 2014), highlighting the importance of long distance advected PM sources, most likely fine PM, on UK health. The $\overline{\text{PM10}}$ from the three sites are as followed: Birmingham Central 20.26 $\mu\text{gm}^{-3}$ ($\log_e[\text{PM10}] = 3.01$), Leicester Central 18.62 $\mu\text{gm}^{-3}$ ($\log_e[\text{PM10}] = 2.92$) and Leamington Spa 18.63 $\mu\text{gm}^{-3}$ ($\log_e[\text{PM10}] = 2.93$). The tri-site $\overline{\text{PM10}}$ is 19.72 $\mu\text{gm}^{-3}$ or $\log_e\overline{\text{PM10}} = 2.98$, resulting in an impact threshold of 29.72 $\mu\text{gm}^{-3}$ or $\log_e[\text{PM10}] = 3.39$.

The concept of a temporal lag between RWB and UK [PM10] is explored during the analysis. The strongest relationships between RWB and MSLP were found if no lag was applied to the MSLP data. When relating MSLP and UK [PM10] a one day lag was found to provide the strongest relationship. The lag accounted for the time taken for European PM10 to advect into the UK and for the UK to subsequently be exposed to a new air mass. Therefore following the diagnosis of RWB, [PM10] was extracted one day subsequent to allow for this lag in changing regimes. In events where RWB was not diagnosed (negative BI values), it was found that a 0-day lag provided the best relationships between RWB and [PM10]. Negative BI values are associated with westerlies entering the UK, providing the most efficient [PM10] removal processes. These removal processes reduce surface [PM10] on timescales of less than a day and hence there is a lag of 0 days between negative BI values and resultant UK [PM10]. Therefore in Sect. 3.1 and 3.2, when for any grid point, a day is not showing RWB, no temporal lag was applied.

## 3    Results

### 3.1    2D spatial relationship between BI and UK PM10

Figure 1 illustrates the relationship between BI magnitude and UK [PM10]. At each grid point the Pearson correlation coefficient between BI at that grid point and PM10 at the central UK Midlands site is shown. A region of positive correlation is found centred over the English channel and two regions of negative correlation are found north and south of this positive correlation region. The region of positive correlation is a result of one of three potential mechanisms, (i) RWB increases the number of high [PM10] events, (ii) RWB reduces the number of low PM10 events, or (iii) a combination of (i) and (ii). By analysing the BI-[PM10] relationship at individual grid points within the region of positive and negative correlation, we can determine which of these mechanisms dominate the relationship.

In order to understand the patterns in Fig. 1, four grid points were selected and labelled GP 1 to 4 on figure 1. GPs 1 and 4 were selected because they lie within two separate regions of negative correlation, while GPs 2 and 3 both lie within the region of positive correlation. Figures 2 (b) and (c) highlight the positive relationships between the BI magnitude and UK [PM10] at GPs 2 and 3, with correlation coefficients (CC's) of 0.42 and 0.32 respectively. The positive relationship is a result of contributions from negative and positive BI values shown as blue and red in Fig. 2. In both cases, the negative BI values





represent a similar positive correlation to that of the overall trend, CC = 0.36 and 0.25 for GPs 2 and 3 respectively. Based upon results of a one tailed t-test, all negative and total correlation coefficients are significantly different from zero at p = 0.01, with none of the positive BI-PM10 relationships showing any significant correlation. For GPs 2 and 3, therefore, variability in the negative BI values dominates the relationship between BI and [PM10]. Subsequently, with negative BI values corresponding to

90   zonal flow, the variability in the magnitude of the zonal flow over these regions explains much of the variability in UK Midland [PM10]. GPs 1 and 4 exhibit a negative correlation between BI and PM10 (CC = -0.28 and -0.34 respectively). The negative BI values in both cases (CC = -0.29 and -0.36 respectively) display near identical relationships with UK [PM10] to the overall trend. Despite the relationships seen between BI and UK Midlands [PM10] being dominated by negative BI values at all four selected regions, positive BI values are associated with elevated [PM10] at GP 2 and 3. At GP 2 and 3, the increased [PM10]

95   associated with positive BI values, indicates that RWB increases the number of high [PM10] events, despite the magnitude of positive BI values not strongly correlating with UK Midlands [PM10].

With the relationship between enhanced zonal flow and low [PM10] dominating the correlations seen in Fig. 2, it can be presumed that the main influence upon this relationship is the EDJ. Figure 1 illustrates a zonal wave pattern of positive correlation between BI and UK [PM10] that is synonymous with the meandering winter EDJ location. Within this region, lie the strongest wind speeds, which are coincident with strong surface wind speeds over the UK, during a zonal synoptic regime. As the EDJ is the dominant feature in the positive correlation between BI and UK [PM10], a positive correlation representing an outline of the meandering winter EDJ location results.

## 3.2   Relationships between Rossby wave breaking subsets and UK PM10

Despite the lack of positive correlation between positive BI values and [PM10], positive BI values are however associated with above average UK [PM10]. In this section we investigate how RWB leads to raised UK [PM10]. Two criteria, described by M12 and M13, are used in this study; the DB and RI indices. Table 1 shows thresholds used to define four RWB types analysed in this study.

For each day on which RWB occurs at a given grid point, the type of RWB is determined from the criteria in Table 1 and the corresponding $\log_e([PM10])$, one day after, is stored for that RWB subset. Following this, UK $\log_e(\overline{[PM10]})$ for each RWB subset is calculated for every grid point. Figure 3 illustrates UK $\log_e(\overline{[PM10]})$ for the four RWB subsets listed in Table 1. Included in Fig. 3 are contoured regions representing grid points where subset $\log_e(\overline{[PM10]})$ is significantly greater than the $\log_e(\overline{[PM10]})$ for the entire dataset (described by Eq. 7), otherwise termed Regions of Influence. For each grid point, the

standard error of the subset $\log_e(\overline{[PM10]})$ was calculated and multiplied by 1.96 to represent a one-tailed 99% confidence interval. Equation 8 describes regions where the subset $\log_e(\overline{[PM10]})$ is significantly less than the dataset $\log_e(\overline{[PM10]})$ and illustrated by a dashed black contour in Fig. 3. Within these equations $\sigma$ represents standard deviation and N represents the number of samples within the subset or dataset.

$$\overline{[PM10]}_{Sub} - \overline{[PM10]}_{All} > 1.96 \sqrt{\left( \frac{\sigma^2_{Sub}}{N_{Sub}} + \frac{\sigma^2_{All}}{N_{All}} \right)} \tag{7}$$



$$\overline{[PM10]}_{All} - \overline{[PM10]}_{Sub} > 1.96 \sqrt{\left( \frac{\sigma_{Sub}^2}{N_{Sub}} + \frac{\sigma_{All}^2}{N_{All}} \right)} \qquad (8)$$

Contoured regions of influence were found outside of the domain shown in Fig. 3, but these incorporated few RWB events randomly coinciding with RWB events within the domain. As these can be assumed coincidental events, the following longi-
tudinal filter has been applied: $277.5^{\circ}\text{E} \leq \text{longitude} \leq 77^{\circ}\text{E}$ in order to focus on regions influential upon UK [PM10]. The longitudinal filter corresponds to spatial bounds that RWB events are identified/ignored when they occur inside/outside of the bounds.

First of note from Fig. 3 are the similarities in location between the cyclonic and anticyclonic RWB regions of influence, with influential regions most prevalent within the Northeast Atlantic/ European region. Climatologically CRWB occurs to the
north of the EDJ, predominantly within the Northwest Atlantic region. However, Fig. 3 shows that the Northeast Atlantic/West European sector, warm and cold CRWB are shown to significantly influence UK [PM10]. Figure 3 (b) shows that warm ACRWB is the most influential RWB sub-category in terms of its regional extent, although warm CRWB exhibits the largest mean $\log_e[PM10]$ magnitudes. The results from Fig. 3 support Fig. 1, as they illustrate the influence of RWB on UK [PM10], predominantly within the same region as the positive correlation in Fig. 1. Following Fig. 3, all RWB sub-categories are
analysed for events occurring within their solid contoured significant regions.

### 3.3  Pressure composites associated with RWB driven UK [PM10] exceedances

M12 and M13 illustrated the climatological pressure response to each of the four RWB subsets in Table 1. M12 show that climatologically, CRWB exhibits a pressure dipole over West Greenland, while ACRWB results in the same dipole centred over West Europe. Fig. 4 illustrates composite MSLP anomalies one day after days incorporating a) ACRWB and b) CRWB
events within their regions of influence and that lead to a UK [PM10] exceedance ($\log_e[PM10] > 3.39$) the following day.

In Fig. 3 for any given day, the PM10 value of that day may be placed into two or more RWB subsets, but for this analysis, only one RWB subset is elected per day based on the following criteria. Firstly the greatest BI value is found in each region of influence. This BI value must correspond to the RWB subset of the region of influence within which it was found, based on the corresponding DB and RI indices for that grid point (eg. Warm ACRWB). A occurrence threshold of 10 positive BI
values, with the same DB and RI metric classifications, has been applied within the region of influence. These criteria have been applied so that the largest RWB events (greatest BI values), must occur within the region of influential RWB for that subset. Furthermore the 10 event threshold, imposes a spatial robustness for that specific RWB subset. If more than one RWB subset has occurred within its region of influence then the RWB subset occurring with the greatest magnitude of overturning (BI value) is selected as the dominant RWB type for that day. Subsequently, only one RWB subset can be selected as the most
influential RWB subset on any given day.

Figures 4 (a) and (b) are constructed using the combined warm and cold ACRWB and CRWB subsets listed in Table 1 and are composed of 50 and 35 contributing days respectively. The figures are similar, with a dominant high MSLP anomaly situated over Scandinavia and a low MSLP anomaly over the Azores. Together the high and low pressure anomalies complete




the anomalous MSLP dipole expected of RWB. In both cases the anomalous MSLP dipole is centred over Northern France,
indicating that the RWB generating this dipole occurred, on average, in this region.

Figures 4 (c) and (d) illustrate the composite MSLP anomaly one day after all CRWB occurrence (regardless of [PM10])
inside the regions of influence, within the Northwest Atlantic and Northeast Atlantic/ European sectors respectively. The
decision to dissect CRWB occurrence into two sectors follows the result that the regions of influence for CRWB have a greater
regional extent in the Northeast Atlantic/ European sector than in the Northwest Atlantic sector (Fig. 3 (c) and (d)). In contrast
it has been highlighted in literature that CRWB is most likely to occur in the Northwest Atlantic sector and to the North of the
EDJ. The purpose of the Fig. 4 (c) and (d) is to determine the region that CRWB occurrence is most likely to lead to [PM10]
exceedances. The longitudinal sectors incorporating these two regions were $225^o$- $329^o$ E longitude for Northwest Atlantic
RWB and $330^o$- $55.5^o$ E longitude for Northeast Atlantic/ European RWB.

The number of events in Fig. 4 (c) and (d) are 52 and 22 in the Northwest Atlantic and Northeast Atlantic/ European sectors
respectively. The greater number of events in Fig. 4 (c) than (d), despite the much smaller regions of influence in the Northeast
Atlantic sector than the Northeast Atlantic/ European sector (Fig. 3 (c) and (d)), highlight the greater density of CRWB events
in the Northwest Atlantic region.

Both Fig. 4 (b) and (d) show similar anomalous MSLP patterns, with an anomalous high MSLP system centred over the
West Scandinavia region. This suggests that CRWB events leading to UK [PM10] exceedances in Fig. 4 (b) are predominantly
Northeast Atlantic/ European CRWB events. Figure 4 (c) however, shows a negative MSLP anomaly stretched across the North
Atlantic towards the UK. This pattern would suggest that Northwest Atlantic CRWB events coincide with conditions prevalent
for UK [PM10] dispersion and removal.

### 3.4 Importance of RWB upon UK [PM10] exceedances

In this section we focus on the RWB events leading to UK [PM10] exceedances. Section 3.1 highlighted that raised UK [PM10]
is associated with positive BI values, showing that RWB has an effect of preventing the strongest westerly zonal winds from
entering the UK region. Figure 3 shows the importance of RWB in significantly raising UK $\overline{\text{PM10}}$, but does not demonstrate
the probability of individual RWB events exceeding a $\log_e$[PM10] threshold of 3.39.

Determining the probability of PM10 exceedance values is perhaps the most important result within this study, as it quantifies
the importance of RWB on [PM10] levels detrimental for human health. As explained in Sect. 2.4 the exceedance value this
study uses is 29.72 $\mu$g m$^{-3}$ ($\log_e$[PM10] = 3.39).

M13, Woollings et al., 2008 and Pelly and Hoskins, 2003 recognised the importance of the persistence of atmospheric block-
ing in influencing synoptic meteorology. As yet, this study has not imposed any persistence criteria on the RWB metrics used.
RWB persistence is expected to prolong the influence of atmospheric blocking and the resultant meteorological patterns. In
the context of this study, it is expected that persistent events would be associated with the most hazardous UK [PM10] events.
Northeast Atlantic/ European RWB leading to atmospheric blocking, has been shown to be associated with anticyclonic con-
ditions influencing the UK (M13), which are favourable for PM10 accumulation (Buchholz et al., 2010). For Fig. 5, persistent



RWB has been defined as RWB following a day of RWB occurrence, both identified within the regions of influence, as defined in Fig. 3.

Figure 5 shows cases when no RWB of any type has been identified within the region of influence for that RWB subset (blue), cases when RWB of the type being analysed has occurred following a day of no RWB (black) and the occurrence of the respective RWB subset following a day of RWB of any type within a region of influence (red).

As for Fig. 4, each day can only be associated with a single RWB subset, which has occurred with the greatest BI magnitude within its region of influence. Furthermore, as in Fig. 4, a threshold of 10 grid points has been used to determine RWB occurrence. Finally, following the finding in Fig. 4 that Northwest Atlantic CRWB has a detrimental influence on meteorology leading to PM10 accumulation, only RWB within the longitudinal bounds of 330 $^o$- 55.5$^o$ E is included in the analysis.

RWB has been analysed in four subcategories; warm, cold, anticyclonic and cyclonic. Values for probability of exceedance are shown on each figure and are calculated as 1-$\alpha$, where $\alpha$ is the probability of occurrence shown in Fig. 5. The probability of exceedance associated with no RWB is 0.129 and all RWB events can be seen to exceed this probability in Fig. 5. From onset (RWB within a region of influence following a day of no RWB within a region of influence) it can be seen that the probability of exceedance is greatest for Cold RWB (0.247) and lowest for Warm RWB (0.207). The greatest probabilities can be seen for continuation events (RWB within a region of influence following a day of any RWB subset within its region of influence). Unlike for onset events, the probability for exceedance for CRWB events (0.379) is greater than that of ACRWB (0.317), with continuous CRWB events representing the greatest exceedance probability in Fig. 5. It is an unexpected result that CRWB, whose climatologically most frequent location is situated in the Northwest Atlantic region, has greater influence upon UK [PM10] than ACRWB, which is most frequent within the Northeast Atlantic/ European sector.

Section 3.3 showed that CRWB is most influential when it occurs within the Northeast Atlantic/ European sector, a region separated from its climatologically most frequent region. Furthermore it was shown that the anomalous MSLP pattern for influential CRWB bears great resemblance to influential ACRWB (Fig. 4 (a) and (b)) and within a similar region of influence (Fig. 3). Section 3.4 quantified the probability of exceedance for CRWB and showed that it has a greater UK [PM10] exceedance probability as a continuous event than ACRWB. Subsequently it is hypothesised that if CRWB is to become influential upon UK PM10, it must be preceded by ACRWB within the Northeast Atlantic/ European region.

### 3.5 Influential CRWB Hypothesis

Climatologically CRWB is most likely to occur within the Northwest Atlantic region due to the cyclonic meridional zonal shear imparted on the background flow by the EDJ to the south (Weijenborg et al., 2012). Despite this, regions of influence within Fig. 3 have shown the largest CRWB regions to be closely located to those of ACRWB, within the Northeast Atlantic/ European region, climatologically south of the EDJ for the winter period (DJF). A hypothesis has been developed that connects the two RWB types (CRWB and ACRWB). It will be shown that the majority of the influence shown in Fig. 3 for Northeast Atlantic/ European CRWB is pre-determined from the occurrence of ACRWB on a preceding day.

Figure 6 shows a schematic, which illustrates how CRWB can become influential, following an ACRWB event. Figure 6 (a) shows the mature stage of an ACRWB event in which the mid-latitude EDJ is displaced to the south of the resultant high





pressure lobe. Subsequently with a more southerly EDJ, colder poleward air is advected into the positive potential vorticity anomaly, such as in Fig. 6 (b), forming an $\Omega$ block. Following the generation of an $\Omega$ block, a cut-off, formed of anomalously negative potential vorticity, warm, equatorward air, may form as in Fig. 6 (c). In Fig. 6, red shading represents regions of

detected ACRWB, while CRWB is represented by blue shading. West (upstream) of the ACRWB event seen in Fig. 6 (b), a CRWB event is detected, which is consistent with the regions of influence seen in Fig. 3 (c) and (d). It is hypothesised that CRWB occurs upstream of the ACRWB due to zonal velocity shear imparted on the background flow by the eastward propagating EDJ approaching the blocking dipole. Subsequently cyclonic vorticity is imparted on the background flow to the north of the EDJ approaching the blocking dipole and CRWB can occur.

### 3.5.1   Dependence of Northeast Atlantic - European CRWB upon the prior occurrence of ACRWB

In Fig. 6, CRWB occurrence within the Northeast Atlantic/ European influential region follows prior ACRWB occurrence in the region. Further analysis has shown that this is the case in 94% Northeast Atlantic/ European CRWB events observed throughout January 1999-December 2008. 35 individual CRWB events were detected throughout the sampling period. A CRWB event was determined when > 10 grid points within the CRWB regions of influence showed CRWB within them. The 10 grid point

threshold helped to remove some smaller transient features. Unlike for Sect. 3.3 and 3.4 no criteria were imposed on the DB and RI metrics of the grid point with greatest BI value. In this section, it is less important to distinguish days influenced more by ACRWB and CRWB events, as Fig. 6 shows that both are influential in the formation of an $\Omega$ block. An ACRWB precursor was detected using the same 10 grid point threshold within ACRWB regions of influence for the four days leading up to and including the day of CRWB occurrence. This indicated that prior ACRWB was likely to be related to the occurrence of CRWB

within this region.

Figure 7 shows the mean count of ACRWB (solid red line) and CRWB (solid blue line) grid points, on the days surrounding a CRWB event. Every event is centred on its peak CRWB grid point count and for each day a 25th and 75th percentile have been included (dashed lines) to test whether the data has been skewed by outliers.

Figure 7 shows the CRWB and ACRWB counts for 94% of Northeast Atlantic/ European CRWB events as defined within

this study. Evident is the influence ACRWB plays upon the occurrence of CRWB, with the mean ACRWB count greater than the climatological average ACRWB count (black line) for the entire period of the CRWB event. The mean ACRWB count peaks one day prior to the CRWB event, with the 25th percentile for this period also exceeding the mean ACRWB count for the dataset (27.66 grid points). With the ACRWB 25th percentile above the black line in Fig. 7 for -1 days, this indicated that the ACRWB precursor is a robust feature for at least 75% of the Northeast Atlantic/ European CRWB events.

None of the Northeast Atlantic/ European CRWB events not associated with an ACRWB precursor led to a [PM10] exceedance on the day following Northeast CRWB detection. For the dataset used to generate Fig. 7, the probability of exceedance is 0.383, which is the greatest value found within this study. UK $\log_e \overline{[PM10]} \pm \sigma$ lagged by one day following day 0 of all CRWB events contributing to Fig. 7 is $3.29 \pm 0.35$. The climatological UK $\log_e \overline{[PM10]} \pm \sigma$ is $2.98 \pm 0.43$ and subsequently the $\overline{[PM10]}$ for the dataset contributing to Fig. 7 is significantly greater to the 99.9th percentile (p = $6.64 \times 10^{-5}$), using the

ANOVA statistical test.



## 4 Discussion

### 4.1 Do large-scale flow patterns influence UK [PM10]?

The relationship between RWB and UK [PM10] has been shown to be one that is dominated by negative BI values (Fig. 1 and 2). Negative BI values represent zonal flow, which provide the most efficient UK PM10 sinks (McGregor and Bamzelis, 1999).

Associated with the strongest zonal flow over the UK is the presence of the EDJ. At times of atmospheric blocking, resulting from RWB over the Northeast Atlantic/ European sector, the EDJ is deflected to the north and/or south of the blocked region (Shutts, 1983), resulting in elevated UK [PM10] (Fig. 2). The correlation tripole in Fig. 1 consisting of positive correlation centred over the English Channel and two negative correlations to the north and south, show a signature of EDJ deflection. The two regions of negative correlation represent regions where the EDJ is deflected to during periods of blocked flow over the UK

and subsequent raised UK [PM10].

The use of RWB as the meteorological diagnostic in this paper differs from classical meteorological diagnostics often used, such as MSLP, wind speed and temperature (McGregor and Bamzelis, 1999; Buchholz et al., 2010; Eder et al., 1994). Justification for the use of the RWB metrics used in this paper can be sought from Woollings et al., 2008. Woollings et al., 2008 showed that RWB can account for the synoptic meteorological variability influencing the UK. The synoptic meteorological

variability influencing the UK can be defined simplistically as zonal flow from the Atlantic influencing the UK or the UK being blocked from this flow. Furthermore RWB contains information relating to the spatial synoptic flow patterns and temporal variability. Temporal variability can be inferred from the upper level field and such as in the case of the Omega block in this study, RWB metrics can tell us about the persistence of such blocked events.

### 4.2 What large scale flow patterns lead to raised UK $\overline{[PM10]}$?

In this study, RWB was shown to significantly (p<0.05) increase UK [PM10], predominantly when it occurred in the Northeast Atlantic/ European region. Weijenborg et al., 2012 showed that ACRWB is most prominent within this region and this study has also shown that ACRWB in this region significantly increases UK [PM10]. Weijenborg et al., 2012 and M13 both highlighted that CRWB is most prominent in the Northwest Atlantic region. M13 highlighted that the dominant effect of North Atlantic CRWB is through the generation of a Northwest Atlantic pressure dipole. This pressure dipole acts to flatten the North Atlantic

EDJ and subsequently displace the EDJ to the south of the UK, reminiscent of the negative phase of the North Atlantic Oscillation. Following M13 showing that North Atlantic CRWB predominantly influences the UK through the removal of the EDJ from the UK, it was expected that CRWB would positively influence UK [PM10] through this mechanism. Furthermore, in Sect. 3.1, it was shown that the dominant mechanism controlling UK [PM10] is the EDJ, which supports the theory that a removal of the EDJ from the UK would influence increased [PM10]. However, rather than Northwest Atlantic CRWB, Sect. 3.2

illustrated that it is predominantly Northeast Atlantic/ European CRWB occurring in a climatologically less frequent location that significantly raises UK [PM10].

Sect. 3.3 showed that both CRWB and ACRWB occurring in their Northeast Atlantic/ European regions of influence and that lead to [PM10] exceedances (>29.72 $\mu$g m$^{-3}$) the following day, shared near identical MSLP responses. This result suggests





that ACRWB and CRWB events were influencing the UK through the same mechanisms and that they are similar events. This
opposes M13, who showed that the most dominant mechanisms influencing UK meteorology for CRWB and ACRWB were
different. As mentioned above, M13 showed that CRWB is influential on Western Europe synoptic meteorology through the
formation of a Northwest Atlantic pressure dipole that acts to flatten the EDJ across the North Atlantic. ACRWB however was
shown to lead to a pressure dipole in the Northeast Atlantic/ European region that blocked Western Europe from the EDJ. The
MSLP response to Northwest Atlantic CRWB in Fig. 4 c), occurring in its regions of influence showed a pattern contrasting
to M13. This MSLP composite shows a negative MSLP anomaly stretching across the North Atlantic over the UK, which
would indicate stronger PM10 removal processes such as precipitation and stronger wind speeds (McGregor and Bamzelis,
1999). Figure 4 d) showed that CRWB occurring in the Northeast Atlantic/ European region bore more resemblance to the
MSLP pattern in Fig. 4 a) of CRWB leading to a [PM10] exceedance one day later. Subsequently, the results from Fig. 4 do
not support the findings of M13 that CRWB is most influential on the UK when it occurs in the Northwest Atlantic region.
Alternatively, as for Fig. 3, Fig. 4 finds that CRWB leads to increased UK [PM10] when it occurs in the Northeast Atlantic
region.

Previous literature has looked at the influence of synoptic scale meteorology on UK [PM10], with high and low frequency
meteorological variability explaining some of the variability present in UK [PM10] datasets. Both McGregor and Bamzelis,
1999 and Buchholz et al., 2010 used clustering techniques to identify low frequency weather regimes that were influential on
UK [PM10]. The primary conclusion was one that identified European anticyclonic regimes as the most influential to raised
UK [PM10], while maritime Atlantic air masses were conducive to low UK [PM10]. The results in this paper support these
findings, as we point to the dominant anomalous MSLP response to RWB that leads to elevated PM10, being an anticyclone
5    over Scandinavia. In the method we used, each day can be classified by a number of continuous metrics, as opposed to clustering
techniques, which use discrete data in the form of regimes to classify each day. The advantage of this method is that no day can
get misrepresented by using only a discrete number of clustered regimes. For example, rare, yet very important events such as
Omega block events are likely to be indistinguishable in a clustering framework.

### 4.3  What large scale flow patterns lead to the greatest probability of exceeding hazardous UK [PM10] limits?

10    Gehring et al., 2013 showed that [PM10] > [PM10] + 10 $\mu g\,m^{-3}$ leads to respiratory mortality in children through long term ex-
posure, but Katsouyanni et al., 2001 also showed that episodic short term exposure of [PM10] > [PM10] + 10 $\mu g\,m^{-3}$ can lead
to an increase in urban mortality rates. Exceedance probability analysis focuses on episodic events above a hazardous [PM10]
threshold, determined following Katsouyanni et al., 2001. The climatological exceedance probability in the UK Midlands is
0.129 and all RWB events led to an increase in this probability associated with their occurrence. The greatest probability of
15    exceeding this studies hazardous [PM10] threshold was associated with a synoptic mechanism identified by an $\Omega$ block (prob-
ability = 0.383). The results in this paper suggest that such events are associated with two dominant mechanisms in raising
UK [PM10]. The first of these is associated with the advection of European PM10, subsequently raising the UK background
[PM10]. The ACRWB shown to precede the $\Omega$ blocking in Fig. 7 is associated with a Scandinavian high pressure system and
easterly advection into the UK, which is thought to facilitate the advection of European PM10 into the UK. Following this ad-





vection, CRWB occurring upstream of the preceding ACRWB event facilitates a persistence of conditions conducive to PM10 accumulation in the UK. This suggests that the occurrence of advection followed by the persistence of UK stagnation leads to the greatest PM10 exceedance probabilities in this study and subsequently the most hazardous [PM10] within the UK. This result will be the subject of future publication, which will help determine the dominant mechanisms by which RWB leads to UK PM10 exceedances.

The increased probability of exceedance associated with continuous events for most RWB subsets, as compared to onset events, (Sect. 3.4) highlights the importance of RWB persistence in generating conditions conducive to hazardous UK [PM10]. Shutts, 1983 detailed a mechanism by which a blocking dipole can be reinforced by eddies propagating from the EDJ circumnavigating the block. The blocking dipole will selectively absorb eddies with positive/negative PV anomalies in its low/ high pressure lobes. This PV reinforcement subsequently allows the block to persist and for conditions conducive to PM10 accumulation in the UK to also persist. Through the $\Omega$ block mechanism theorised in Fig. 6, it is suggested that eddy absorption allows the blocking dipole to persist as the EDJ circumnavigates the blocking dipole to the north.

The analysis of synoptic weather patterns suggests that the advection of European PM10 heavily contributes to the measured [PM10] at the UK Midlands tri-site. Despite this, no source attribution study was undertaken throughout this analysis and subsequently no quantitative evaluation of the contribution of advected PM10 has been made. A potential follow-up study would be to quantitatively analyse the contribution of advected European PM10 in such RWB events, thereby attributing the sources contributing to hazardous UK PM10 events. Furthermore as RWB has been shown to increase the probability of exceeding a hazardous PM10 threshold limit, the changes in RWB frequency and potency must be understood through climate predictions. Climate predictions alongside quantitative PM10 source attribution studies and this work highlighting the hazard associated with RWB events, will allow for mitigation measures to be implemented in sufficient time to alleviate the potential hazard PM10 poses to the UK population, through the mechanisms described in this paper.

## 5 Limitations and variability

### 5.1 Relating pressure and $\theta - 2\text{PVU}$ fields

This analysis looks at large scale dynamical features affecting regional scale UK [PM10] and in this relationship there is an unconstrained intermediate. This intermediate is the pressure dipole response to RWB, which is not consistent in proximity with the centre of overturning (maximum BI value) in a RWB event. Previous studies have shown the anomalous high pressure lobe to be located to the north of the RWB event, also found within this study. Despite latitudinal consistency, longitudinal differences in the distance between the centre of overturning and the local high pressure maxima exist. Subsequently variability exists within the results in Fig. 3, as UK [PM10] is not solely dependent upon the location of overturning. The relationship between location of BI maxima and high pressure maximum remains unconstrained and therefore inherent uncertainty of this relationship exists.



## 5.2 Observational tri-site

The observational sites used within this study are all classified as urban background sites, therefore they will be influenced by their urban environments independently. Subsequently there will be periods in the data that local influences out-weigh the effects of large scale episodes. Furthermore the tri-site is used to represent UK from its most central point and in doing this, it is likely that some events, particularly some continental advection events affecting the South East corner of the UK will not be recorded. The extent to which three sites can represent the entire country is limited, but due to data availability, it is not possible to fully represent the entirety of the UK without large scale data interpolation methods, which are heavily influenced by the local influences of all sites used. The analysis within this paper has been repeated using data from both a rural site at Harwell, Oxfordshire and Southampton, Hampshire.

The data from Southampton showed regions of influence with reduced magnitudes of extent for all RWB subsets, but especially for the two ACRWB subsets. It is hypothesised that ACRWB facilitates the advection of European PM10, driven by a Scandinavian anticyclone. Subsequently, in Southampton, where European pollution sources more regularly have a greater influence (Malcolm et al., 2000) than in the UK Midlands, the advection of European PM10 is not associated with as great a [PM10] increase. Despite the reduced magnitudes, the regions of influence were spatially co-located with those illustrated in Fig. 3. Conclusions gathered from the analysis of exceedance probabilities did not change greatly. The exceedance probability for all RWB was elevated above the exceedance probability with no RWB (0.169). Continuous CRWB events led to the greatest exceedance probability (0.659) of the RWB subsets in Fig. 5, which was much greater than the exceedance probability for continuous ACRWB (0.323). Analysis of $\Omega$ blocks showed that these events led to the second greatest probability of exceedance in Southampton (0.604). However, as for the UK Midlands PM10 dataset, the probability of exceedance for $\Omega$ block events is similar to that for continuous CRWB events, as they comprise of many of the same events. Despite this and due to the reduced ACRWB regions of influence, only 11 out of 17 Northeast Atlantic CRWB events were associated with an ACRWB precursor.

Data from the rural monitoring site in Harwell showed little difference in the shape or locations of the regions of influence for all RWB subsets, when compared with the UK Midlands dataset. Furthermore the climatological exceedance probability was slightly reduced (0.082), as the 10 $\mu\mathrm{g\,m^{-3}}$ increment used to define an exceedance, represented a much greater proportion of the sites' $\overline{\mathrm{[PM10]}}$. Continuous CRWB events led to the greatest exceedance probability (0.368) of the RWB subsets illustrated in Fig. 5 as before. As for the two urban background sites, $\Omega$ block events led to the greatest overall exceedance probabilities (0.381), where 30/33 Northeast Atlantic/ European CRWB events were identified as $\Omega$ block events.

## 6 Conclusions

The influence that Rossby Wave Breaking (RWB) has on UK PM10 concentration ([PM10]) has been analysed, with significant relationships present. Positive correlations exist within the Northeast Atlantic/ European region, between UK [PM10] and the Blocking Index, a metric used to diagnose RWB. Primarily, the UK experiences either zonal or blocked flow, with the latter attributable in winter months, to RWB. Zonal flow is associated with the strongest PM10 loss mechanisms and it is the removal



of zonal flow from the UK that dominates the positive correlation between RWB and UK [PM10]. The strongest zonal flow
speeds and subsequent greatest PM10 loss mechanisms are associated with the Eddy driven jet (EDJ). The EDJ has shown to
be the dominant mechanism in the relationship between RWB and UK PM10.

Analysis was undertaken to determine whether RWB can aid in the accumulation of UK [PM10] above background con-
centrations. Regions of influence highlighted that RWB occurring in the Northeast Atlantic/ European region resulted in a
significant ($p<0.05$) increase in UK [PM10] above climatological levels. Anticyclonic RWB (ACRWB) was seen to signifi-
cantly raise mean [PM10] ($\overline{[\text{PM10}]}$) when it occurred in the Northeast Atlantic/ European region. Additionally Cyclonic RWB
(CRWB) significantly raised UK PM10 when it occurred in a similar region, despite being climatologically uncommon in
this region, due to the anticyclonic nature of background flow on the equatorial side of the EDJ. The MSLP response to both
CRWB and ACRWB, resulting in PM10 exceedances is dominated by an anomalous anticyclone, centred over Scandinavia.
This results in easterlies into the UK and the advection of European PM.

As RWB events are deviations from the synoptic climatological mean state, the probability of exceeding an episodic [PM10]
threshold value was used to evaluate the anthropogenic health hazard associated with PM10. All RWB subsets led to an in-
crease in exceedance probability above climatology, when they occur within constrained regions of influence. The occurrence
of CRWB led to the greatest exceedance probability of any RWB subset. This was an unexpected result, as ACRWB is clima-
tologically more frequent in the Northeast Atlantic/ European region and subsequently more influential on UK meteorology
than CRWB.

A mechanism was hypothesised, explaining the occurrence of CRWB within a region separate from its climatological most
frequent region. The mechanism depended on the prior occurrence of ACRWB within the Northeast Atlantic/ European region
and was characterised by the formation of an $\Omega$ block. The persistence of these events led to the probability of exceedance in
such cases, exceeding that of all other RWB subsets, with the added influence of UK stagnation influencing raised UK [PM10].
The mechanism by which these events influence UK [PM10] is through the initial advection of European PM10 into the
UK through ACRWB within the Northeast Atlantic/ European region. Subsequently, a prolonged period of stagnation allows
the accumulation of local PM10 sources to increase UK [PM10] further. The occurrence of advection and the persistence of
stagnation is hypothesised to lead to the greatest UK [PM10].

Further analysis has been suggested to quantitatively evaluate the sources of PM10 throughout such events. Furthermore
analysis must be undertaken regarding the frequency of such events in a changing climate. Once source regions and the fre-
quency of PM10 exceedances, resulting from RWB in a future climate are known, mitigation steps can be made (if required)
to alleviate any potential increase in RWB frequency or potency in a changing climate, through local or continental emis-
sion strategies. Without a full understanding of the relationship between meteorology, hazardous pollutant events and climate
responses, the opportunity to implement effective mitigation steps now, may be lost.

### Acknowledgments

Chris Webber was funded by a NERC PhD Studentship



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

**Table 1.** Defining Rossby wave breaking (RWB) types using thresholds of the direction of blocking (DB) and relative influence of air masses (RI) indices. Any RWB with DB or RI values of $-0.2 \leq x \leq 0.2 \, \mathrm{K\,km^{-1}}$ are determined as unclassified following Masato et al., 2012.

| Rossby Wave Breaking Type | Blocking Index (BI) $\mathrm{K\,km^{-1}}$ | Direction of Blocking Index (DB) $\mathrm{K\,km^{-1}}$ | Relative Influence of Air Masses Index (RI) $\mathrm{K\,km^{-1}}$ |
|---|---|---|---|
| Warm Anticyclonic | > 0 | > 0.2 | > 0.2 |
| Cold Anticyclonic | > 0 | > 0.2 | < -0.2 |
| Warm Cyclonic | > 0 | < -0.2 | > 0.2 |
| Cold Cyclonic | > 0 | < -0.2 | < -0.2 |





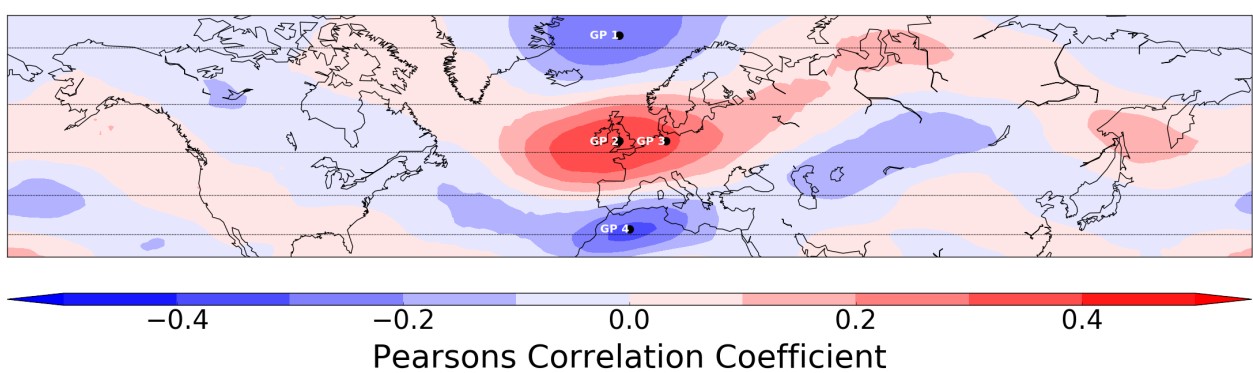

**Figure 1.** Pearsons correlation coefficient between Blocking Index magnitude ($\mathrm{K km^{-1}}$) and UK [PM10] ($\mu\mathrm{gm^{-3}}$) in the UK Midlands region. All data was taken for the winter months (DJF) between January 1999 and December 2008. PM10 data following a positive/negative BI value was lagged by 1/0 days respectively. Grid points selected for further analysis are labelled GP 1 to 4.




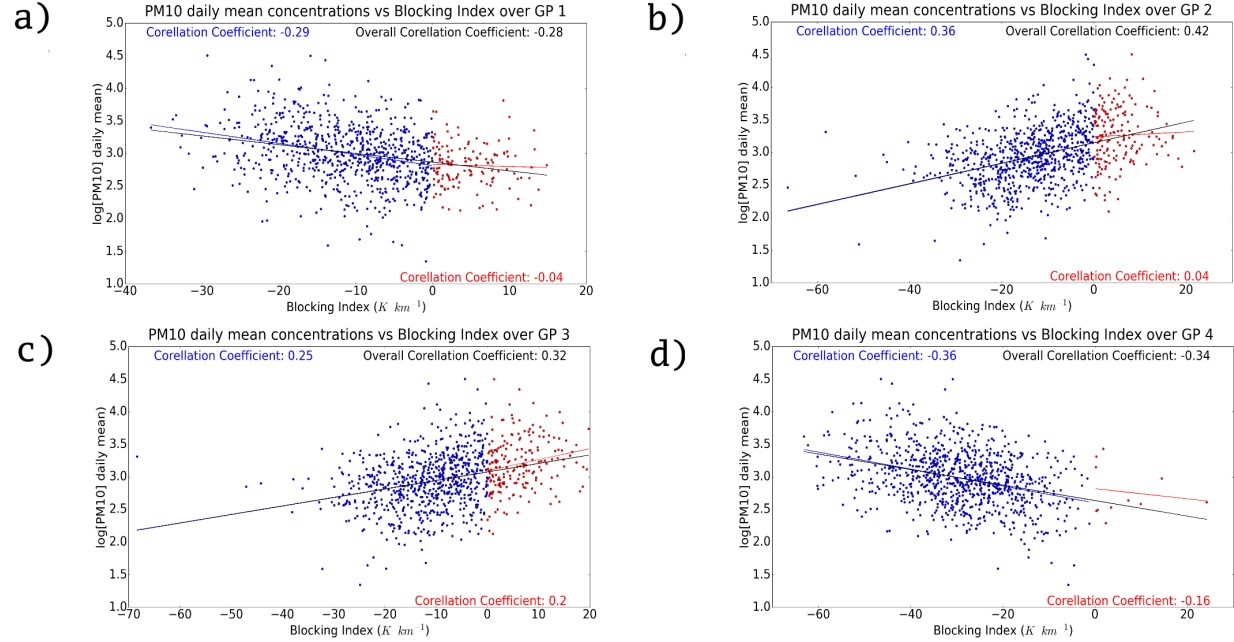

**Figure 2.** Scatter plots illustrating the relationship between Blocking Index magnitude (BI) ($\mathrm{Kkm}^{-1}$) and UK [PM10] ($\mu\mathrm{gm}^{-3}$) for four regions labelled GP 1, 2, 3 and 4 on Fig. 1 (Labelled a, b, c and d on Fig. 2). The blue/red colour corresponds to negative/positive BI values respectively and the fitted trends are linear least-squares trends. Data was obtained for DJF January 1999 - December 2008 and [PM10] lagged by 1/0 days for positive/negative BI values respectively.

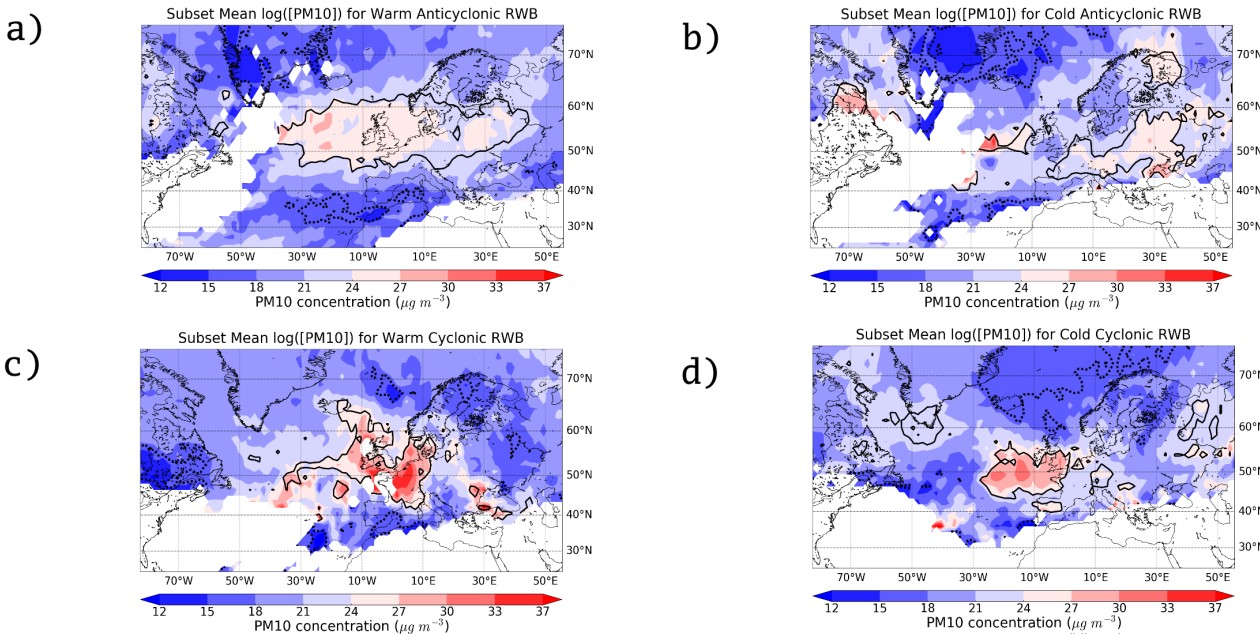

**Figure 3.** One day lagged UK Midlands mean [PM10] subset ($\overline{[\mathrm{PM10}]}_{\mathrm{Subset}}$) for each point within a gridded region for : a) warm anticyclonic, b) cold anticyclonic, c) warm cyclonic and d) cold cyclonic Rossby Wave Breaking events. Solid/dashed contours indicate regions where the $\overline{[\mathrm{PM10}]}_{\mathrm{Subset}}$ is significantly higher/lower than the mean $\log_e([\mathrm{PM10}])$ for the entire dataset respectively. White grid points represent points where RWB has occurred on less than 2 occasions throughout DJF January 1999 - December 2008. Dataset mean [PM10] = $21.69\,\mu\mathrm{g\,m^{-3}}$.



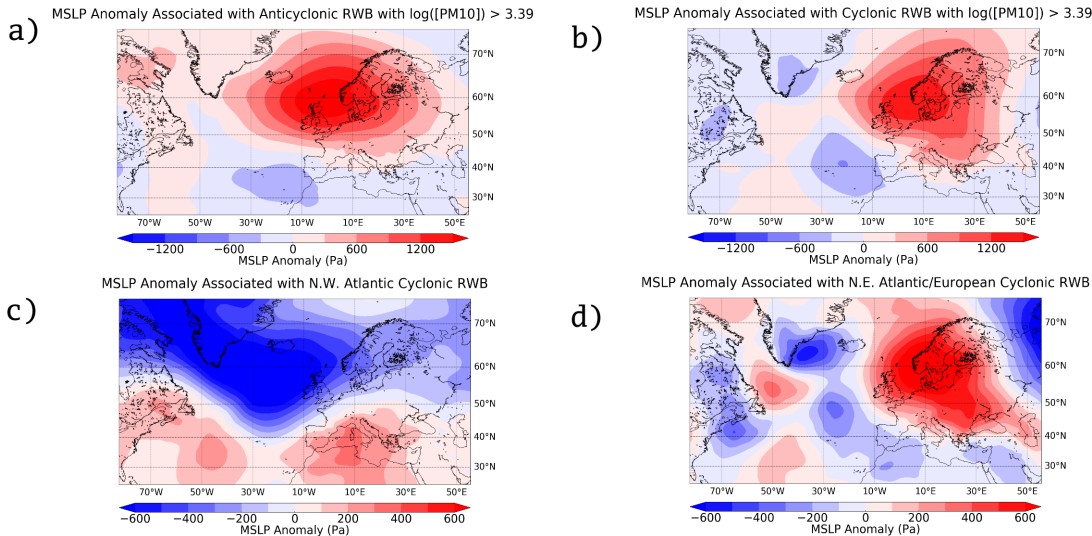

**Figure 4.** a) and b) Composite MSLP anomaly on the day following days incorporating; anticyclonic RWB and cyclonic RWB within the regions of predefined influence (see Fig. 3) respectively and on the following day $\log_e$[PM10] > 3.39. c) and d) MSLP anomaly on the day following days incorporating cyclonic RWB within the regions of predefined influence in the Northwest Atlantic and Northeast Atlantic/ European sectors respectively, regardless of [PM10]. Pressure data was taken for the winter months (DJF) between January 1999 and December 2008.





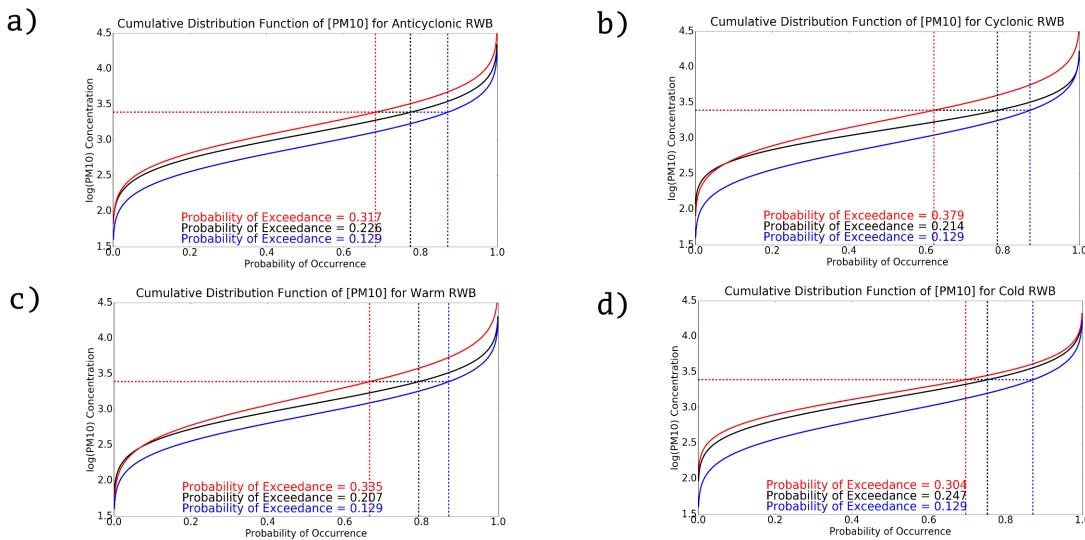

**Figure 5.** Cumulative distribution functions (CDF) for UK $\log_e$ [PM10] and a) Anticyclonic, b) Cyclonic, c) warm and d) cold RWB. The red line illustrates the CDF for each RWB subset following a day with RWB of any type within the respective region of influence (continuous event). The black line illustrates the CDF for each RWB subset following a day with no RWB of any type (onset event) and the blue represents [PM10] associated with no RWB. Data was obtained for DJF January 1999 - December 2008 and [PM10] is lagged by one day following detection of RWB. Values shown for probability of exceedance are calculated as 1-$\alpha$, where $\alpha$ is the probability of occurrence.




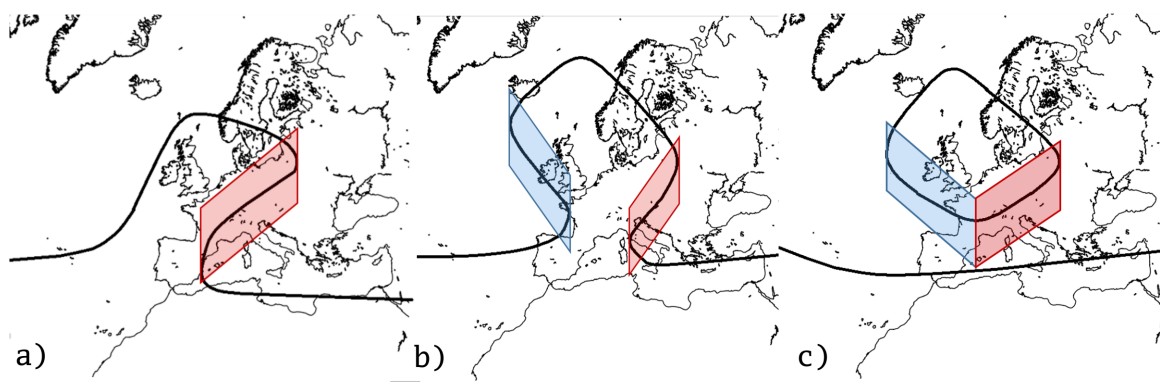

**Figure 6.** a) Schematic of a mature Warm-Anticyclonic Rossby wave breaking event on the dynamical tropopause, with a region of positive BI Index and anticyclonic DB Index values shaded red. The black contour portrays an arbitrary potential temperature contour representing the intermediate between anomalously cold and warm air masses. b) Schematic of an $\Omega$ block with an additional Cold-Cyclonic Rossby wave breaking event (blue) following a). c) Schematic of a final stage warm $\theta$ cut-off.





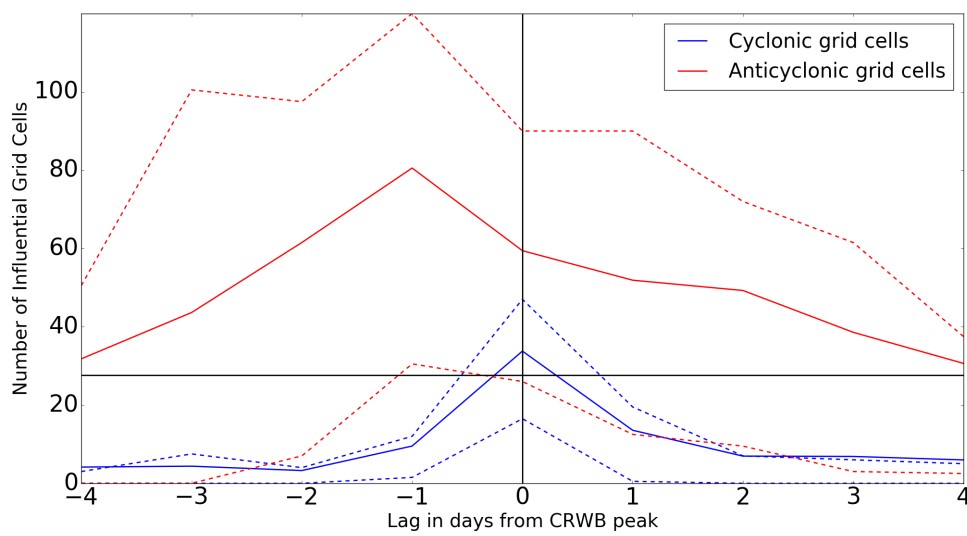

**Figure 7.** Grid points displaying RWB of (Anti)cyclonic tendency in (red)blue with time in days relative to the peak of Northeast Atlantic/ European CRWB events. Solid lines represent the mean grid point count, while the dashed lines represent the 25th and 75th percentile of counts for all events. In the formation of an Omega block an ACRWB precursor is required. Of the 35 CRWB events detected within the Northeast Atlantic/ European sector throughout January 1999 - December 2008, 33 fulfilled the criteria of prior ACRWB occurrence. All CRWB events displayed in Fig. 7 include an ACRWB precursor within the four days leading up to and including the CRWB event. The black horizontal line represents the mean ACRWB count (30.07 grid points) throughout the dataset.