# Peer review of "The Dynamical Impact of Rossby Wave Breaking upon UK PM10 Concentration"

_Atmospheric Chemistry and Physics, 2016_

## Referee Comment (RC1) · Anonymous Referee #1 · 1 Sep 2016

The manuscript is describing in detail the influence of Rossby Wave Breaking (RWB) on PM10 daily average concentrations over the central UK. The meteorological data, from which the blocking index were calculated, are obtained from the ECMWF ERA-Interim, while data of PM10 are from station observations for the period 1999-2008. The manuscript is well written, the analysis is well described and sound. I believe that this manuscript should be published on ACP.

I would however appreciate if the authors could clarify few issues and correct minor inconsistencies, so to improve the manuscript even further.

**Major comments:**

1. I appreciate the work done to create a "super-site". However, few additional information would indeed help. A correlation plot between the 3 different sites before and after the "tendency outliers" removal would be great to have, and possibly these should be added on the electronic supplement. I would be very interest to see if the 3 sites do correlates at least after your corrections. If this is not the case, probably the super-site estimations are without any real meaning, as mainly influenced by local emissions.

2. I was not really able to find the information regarding threshold in PM10 concentration in the manuscript of Gehring et al. (2013), i.e. concentration value below which PM10 does not have any health effect. To my knowledge, this is matter of debate, and the World Health Organization writes that: "Small particulate pollution have health impacts even at very low concentrations indeed no threshold has been identified below which no damage to health is observed." (see http://www.who.int/mediacentre/factsheets/fs313/en/). Therefore I do not argue with the thresholds selected in section 2.2, but I would remove the health considerations. See also the discussion in Burnett, R. T. et al. (2014)

**Minor comments:**

**Page 1, line 2** : please mention that only daily PM10 values are used.

**Page 2, line 53** : I am puzzled with the "climatological mean sea level pressure", defined here. I think that "climatological" refer to 30 years average. Maybe the authors refers to daily mean sea level pressure as produced by the ECMWF era-interim product.

**Page 2, line 54** : Please specify that theta is potential temperature.

**Page 5, line 62** : You refer to a correlation between RWB and [PM10]. From your work I understood that the BI was used to represent RWB. I would therefore either

change RWB with BI or specify that the time lag is estimated between BI and [PM10]

**Page 6, line 13** : There is an inconsistencies between figures and text. In Fig.3 the caption mention only log[PM10] and not $log_e$ (or ln). However, in the color bar, the absolute values are used (i.e. wihout any logarithmic calculation). Why not using "ln"(natural logarithm) in the entire manuscript? Additionally, in Fig.3 you mentioned that the solid/dashed lines represent regions where the subset average of daily PM10 are higher/lower than the mean of $log_e([PM10])$. Should not be the logarithm in both cases? Alternative you could remove the logarithm in the second case. The text read as the solid dashes are all the regions with the average of subset of daily PM10 higher than 2.3 $\mu g/m^3$ !!

**Page 8, line 74** : I must disagree, as Sect. 3.1 only showed that without RWB events (i.e. BI lower than 0), PM10 is more effectively removed (transported) due to the zonal flow, while Sect. 3.2-3.3 showed which kind of special RWB could lead to increase PM10.

**Page 9, line 15** : I find this paragraph extremely difficult to understand, and I had to read it many times to guess what the author means. Would it be possible to reformulate it?

**Page 10, line 55** : Please explain ANOVA acronyms (ANalysis Of VAriance)

**Page 12, line 10** : I think you mean [PM10] > $[\overline{PM10}]$ + 10 $\mu gm^{-3}$. However, please check the major comments on this issue.

**Discussion** :

Would be good for the author to extend the outlook in their discussion :

[Figure]

1. From this work I could also conclude that this should be valid not only for PM10 but also for PM2.5, which are by far mor influenced by transport due to their lower settling velocities. Do the authors have any opinion on that?

2. Could the author extend the manuscript or discuss the absolute frequency of European CRWB ?

**Reference:**

Burnett, R. T. et al. An integrated risk function for estimating the Global Burden of Disease attributable to ambient fine particulate matter exposure. Environ. Health Perspect. 122, 397–403 (2014).

---

## Referee Comment (RC2) · Anonymous Referee #2 · 5 Sep 2016

**1   Overview**

This paper presents a new set of insights into the links between pollution events in the UK and synoptic weather. Its novelty primarily lies in the use of 2D blocking diagnostics and their links with PM10 in the English Midlands. The conclusions reached are substantial and the scientific methods used are valid. There is however, some scope for clarifying the presentation, especially for an audience who may not have as much familiarity with dynamical meteorology as the authors.

With some changes, as described below, this paper is suitable for publication in ACP.

[Figure]

**2 Specific comments**

**2.1 Abstract**

Please include the probability of PM10 excedences both for days without RWB and for those conditions most likely to lead to an episode. Also make it clear that an exceedence when there is RWB is 3 time more likely than periods without RWB (it is not clear currently what it is 3 times more likely than).

**2.2 Introduction**

It would be helpful to expand the description of what is meant by Rossby Wave breaking here. Start by a quick reminder of what a Rossby wave is and then give a bit more explanation of what is meant by large scale overturning (and that it is not overturning in the vertical!). A figure similar to Figure 2 of Masato et al 2012 would be useful to better orient the reader and also to help explain the diagnostics in section 2.

When discussing the way that high pressure influences concentrations, is not the suppression of vertical mixing by large scale subsidence also a factor which may play a role?

**2.3 Section 2.1**

You state the ERA-Interim data has been temporally filtered, but has it also had a running mean in longitude applied as described in M11 and M13? If so please state this, if not explain why.

**2.4 Section 2.2**

I found the section explaining the exceedance threshold confusing. Please remove the first sentence and start the paragraph with "In this study PM10 exceedances are defined using a threshold based on the results of the European Study of Cohorts for Air Pollution effects (ESCAPE) . . ."

At the end of the paragraph replace the last part of the final sentence (after =2.98) with Therefore we use a threshold for daily mean [PM10] of 29.72 ug/m3 or loge[PM10]=3.39 to define an exceedance.

It would also be useful to put this threshold in context by comparing to EU air quality standards and the UK DAQI for example.

**2.5 Section 3.2**

It would be useful in paragraph 1 to summarise in the text what the four RWB types are.

The final sentence of the final paragraph is crucial to understanding the following sections of the paper and needs to be more explicit. Something like "Therefore in all subsequent analyses in this paper, we select only those RWB events occurring in the respective solid contoured regions of influence as shown in figure 3."

**2.6 Section 3.3**

The sentence starting: "Fig 4 illustrates . . . that lead to a UK PM10 exceedance the following day" is unclear. Is the PM10 exceedance one day after the MSLP anomaly (2 days after the RWB event) or one day after the RWB event? Please clarify. If it is the former, please give more information on why a lag between PMSL and PM10 was

used.

**2.7 Section 3.4**

Please include a new paragraph after paragraph 2 to introduce the CDFs here. Describe figure 5 focussing first on the blue and black lines. Then move on to the importance of persistence. It might even be useful to have 2 separate sub-sub-sections for these.

This analysis only covers events with a persistence of 1 day. Have longer periods of persistence been considered or are there too few of these for statistical significance?

**2.8 Section 5.2**

Would it be appropriate to present the figures from the other observations sites as supplementary material?

**3  Technical corrections**

- P5, L59 followed -> follows
- P7L49 Subsequently -> This ensures that
- P8, L71 prevalent -> favourable
- P9, L11 Subsequently -> therefore
- P9, L11. Is pre-determined from -> depends upon
- P12, L104. Buchholz et al – missing reference

- P12 L10 missing over bar [PM10]]

- P14, L62. Subsequently -> Consequently

---

## Referee Comment (RC3) · Anonymous Referee #3 · 8 Sep 2016

Review for
"The Dynamical Impact of RossbyWave Breaking upon UK PM10 Concentration"
by Webber et al.
* * *
**Synopsis:**

Webber et al. discuss in their study if and how the synoptic-scale weather situation influences the UK PM10 concentrations. This is done based on ananalysis of Rossby-wave breaking (RWB), which is further categorized into cyclonic and anticyclonic RWB and into the air masses associated with it. The manuscript is already quite clearly written, the methods used are well described and suitable to support the scientific findings. Further, the results are of interest to the readership of ACP and I, hence, can recommend publication if some minor concerns and clarifications are handled.

**Minor Concerns:**

**-P2,L54-55:** "The pressure dipole results from the meridional advection of upper level air masses with anomalous potential vorticity (PV) characteristics." I think that the link between upper-level PV and the pressure dipole is not immediately clear. A clarifying sentence might help.

**- Section 2.1**: Please move the formulas to the place in the text where they are referred to. At the moment, for instance, formula (6) is appended at the end of the section, but discussed on top of page 4.

- At **P4,L41** the three measurement sites, being classified by DEFRA as urban background sites,  are 'justified' (motivated) by the fact that the majority of UK's population live in urban areas. Further down (L44-48) it is discussed that the sites are influenced by 'urban' activities, and that, therefore, the three stations are combined to remove local spikes. How does this fit together with the motivation? By the way: What is DEFRA?

**- P5,L53 and L61**: "A daily mean [PM10] ([PM10]) exceedance has been defined in this study, when [PM10] exceeds the threshold of 29.72 $\mu gm_{-3}$ ($\log_e$[PM10] = 3.39)" & "The tri-site [PM10] is 19.72 $\mu gm_{-3}$ or $\log_e$[PM10] = 2.98, resulting in an impact threshold of 29.72 $\mu gm_{-3}$ or $\log_e$[PM10] = 3.39": Repetitive?! Further, why does this, in the 2nd sentence, ***result*** in an impact value?

- **P5,L62-70**: Here the time-lag issue is discussed. It is argued that a time lag between [PM10] and RWB makes sense. But, no time lag is used for negative BI and [PM10]. Finally, the whole paragraph starts with the finding that best correlations result if no time lag is used between RWB and MSLP. The reader can easily get lost in these many different cases! A little remedy could be if the link between MSLP and RWB is not discussed. To me, it sounds rather obvious that the best correlation occurs if no time lag is used. And, by the way, I don't see a need to motivate the RWB time lag by a corresponding MSLP time lag. In short: Simplify the paragraph a little!

- **Section 3:** This section contains the main results from the study. All in all, the discussion is clear and the results are well supported by the data. However, while reading from subsection 3.1 to 3.2, to 3.3 and finally 3.5 I got a little lost. Many aspects of the link between [PM10] and RWB, positive and negative BI, CRWB and ACRWB, the exceedances of [PM10]... are discussed. I think it would be great to start the whole section 3 with a (rather short) introductory paragraph that, from the beginning, tells the reader where the journey will go to. In short: Give the reader some guidance what he can expect from this section and how the different subsections are connected.

- **P6,L20**: move formula (7) and (8) to the place in the text where they are referred to.

- **Figure 2 and corresponding text**: In panel b) and c) there are some data points at rather low BI values. I wonder whether these data points, with a considerably leverage, influence the overall fit of the least-square fit? How does the correlation change if these points are omitted? I am also not perfectly convinced that it is reasonable to look at the BI>0 points only (red points and curve fitting) and to deduce that, for instance in panel b), the BI>0 has no impact on [PM10]. Finally, it might be better to use for all four panels the same range for the x axes. This would allow the different locations (GP1-4) to more easily be compared.

- **P7,L24-25**: "the following longitudinal filter has been applied: 277.5oE < longitude < 77oE in order to focus on regions influential upon UK [PM10]." I do not clearly understand what you mean with longitudinal filter? Do you simply neglect all RWB events in this domain?

- **P7,L30-31:** Incomplete sentence?!

- **P10,L33-35**: Repetitive?! Is it the same 10-grid point criterion used before? If so, I would prefer if this criterion is introduced only once in the text.

- **P12, L10: "**[PM10] > [PM10] + 10 µgm−3" Unclear!

- **P12,L15:** "The greatest probability of exceeding this studies hazardous [PM10] threshold was associated with a synoptic mechanism identified by an block (probability = 0.383)" Complicated sentence! Please rephrase.

---

## Author Comment (AC1) · 4 Nov 2016

**Response to the major comments made by Anonymous Referee 1**

**1.** I appreciate the work done to create a "super site". However, few additional information would indeed help. A correlation plot between the 3 different sites before and after the "tendency outliers" removal would be great to have, and possibly these should be added to the electronic supplement. I would be very interested to see if the 3 sites do correlate at least after your corrections. If this is not the case, probably the super-site estimations are without any real meaning, as mainly influenced by local emissions.

*Following the advice to clarify the data verification steps taken to generate a "super-site", further analysis was undertaken. Pearson's correlation coefficients were determined between each constituent PM10 dataset, prior to and following the data verifica-*

[Figure]

*tion step of removing tendency outliers. Prior to the data verification step the Pearson's correlation coefficient values between each [PM10] dataset varied between 0.73 and 0.86, with the lowest correlation coefficient seen between the Leamington Spa and Birmingham Central [PM10] datasets. Following the data validation stage, the Pearson's correlation coefficient values varied between 0.86 and 0.87.*

*These results are referred to in the main text on P4 L50:*

*"Additional analysis (not shown) has found that the data validation step of removing PM10 spikes has improved the Pearson's correlation coefficient between each original PM10 dataset. In the original [PM10] datasets, the correlation coefficients between the three observational PM10 sites varied between 0.73 and 0.86. Following the data validation step, the correlation coefficients varied between 0.86 and 0.87. "*

**2.** I was not really able to find the information regarding threshold in PM10 concentration in the manuscript of Gehring et al. (2013), i.e. concentration value below which PM10 does not have any health effect. To my knowledge, this is matter of debate, and the World Health Organisation writes that: "Small particulate pollution have health impacts even at very low concentrations indeed no threshold has been identified below which no damage to health is observed." (see http://www.who.int/mediacentre/factsheets/fs313/en/). Therefore I do not argue with the thresholds selected in section 2.2, but I would remove the health considerations. See also the discussion in Burnett, R. T. et al. (2014).

**Reference**

Burnett, R. T. et al. An integrated risk function for estimating the Global Burden of Disease to ambient fine particulate matter exposure. Environ. Health Perspect. 122, 397-403 (2014).

*We agree that PM10 below the threshold used has not been shown to be dissociated with detrimental health effects. The threshold used is intended to highlight a thresh-*

[Figure]

*old that concentrations above which have been shown to lead to significant increases in detrimental human health effects. Perhaps a better argument would be to refer to Katsouyanni et al. (2001) instead. Katsouyanni et al. (2001) find a statistically significant increase in mortality rates, associated with episodic PM10 increases of 10 $\mu g$ $m^{-3}$ above the background mean concentration.*

*The justification for the threshold has remained health motivated. We have instead used the results found by Katsouyanni et al. (2001) in the APHEA2 Project to motivate the hazardous UK Midlands [PM10] threshold.*

*The point that PM10 is hazardous to human health at low concentrations and a reference supporting this have been added to the text (P5L58).*

*"It is worth noting that while significant increases in detrimental health impacts have been shown for the threshold used in this study, detrimental health effects have been shown to occur at lower ambient [PM10] (Brook et al., 2010)."*

*An additional reference was included:*

*Brook, R. D., Rajagopalan, S., Pope, C. A. III, Brook, J. R., Bhatnagar, A., Diez-Roux, A. V., Holguin, F., Hong, Y., Luepker, R. V., Mittleman, M. A. and Peters, A.: Particulate matter air pollution and cardiovascular disease an update to the scientific statement from the American Heart Association. Circulation, 121, 2331-2378, 2010.*

**Response to the minor comments made by Anonymous Referee 1**

**3. Page 1, line 2:** please mention that only daily PM10 values are used

*This clarification was added to the abstract at this point.*

**4. Page 2, line 53:** I am puzzled with the "climatological mean sea level pressure", defined here. I think that "climatological" refer to 30 years average. Maybe the authors refers to daily mean sea level pressure as produced by the ECMWF era-interim product.

*The terminology has been changed, as we have not used 30 years of MSLP to obtain our averages. The term "daily mean" has replaced the term "climatology".*

**5. Page 2, line 54:** Please specify that theta is potential temperature.

*Theta has been replaced by potential temperature on Page 2, line 46*

*$\Theta$ has been replaced by Potential temperature on Page 2, line 47.*

*The text "The $\Theta$-2PVU surface" has been replaced by "$\Theta$-2PVU" on Page 2, line 48.*

**6. Page 5, line 62:** You refer to a correlation between RWB and [PM10]. From your work I understood that the BI was used to represent RWB. I would therefore either change RWB with BI or specify that the time lag is estimated between BI and [PM10].

*The text has been edited, so that RWB has been replaced with BI.*

**7. Page 6, line 74:** There is an inconsistency between figures and text. In Fig. 3 the caption mentions only log[PM10] and not loge (or ln). However, in the color bar, the absolute values are used (i.e. without any logarithmic calculation). Why not use "ln" (natural logarithm) in the entire manuscript? Additionally, in Fig. 3 you mentioned that the solid/ dashed lines represent regions where the subset average of daily PM10 are higher/ lower than the mean of $\log_e$([PM10]). Should not be the logarithm in both cases? Alternatively you could remove the logarithm in the second case. The text read as the solid dashes are all the region with the average subset of daily PM10 higher than 2.3 $\mu$g m$^{-3}$.

*We are unable to find log[PM10] in the caption of Fig. 3, however there does exist an inconsistency between $\log_e$[PM10] in the text and [PM10] in the figure (colour bar). This inconsistency has been corrected, so that the colour bar now represents ln[PM10] values. Furthermore all instances of $\log_e$([PM10]) throughout the text have been changed to read ln[PM10].*

*The text in the caption to Fig. 3 describing the solid contours has been altered. The*

*solid contours have been defined to, "constrain regions where mean ln[PM10] for each grid point is significantly greater than the entire dataset ln[PM10] mean (p<0.01)". This is consistent with the description in the main text.*

**8. Page 8, line 74:** I just disagree, as Sect. 3.1 only showed that without RWB events (i.e. BI lower than 0), PM10 is more effectively removed (transported due to the zonal flow, while Sect. 3.2-3.3 showed which kind of special RWB could lead to increased [PM10].

*The line in question refers to raised UK [PM10] associated with positive BI values. This is shown, in Fig. 2 b) and c) for GP 2 and 3 respectively. Due to the positive correlation that exists between BI and UK [PM10] in Fig. 2 b) and c), it can be stated that increased BI values (positive BI values) are associated with increased UK [PM10].*

**9. Page 9, line 15:** I find this paragraph extremely difficult to understand, and I had to read it many times to guess what the author means. Would it be possible to reformulate it?

*The following text has replaced the paragraph originally beginning on P9L14:*

*"The mechanism primarily dictating the direction of RWB is the meridional hear of the zonal wind, which is imparted on the background atmospheric flow by the EDJ. To the north/ south of the EDJ, a cyclonic/ anticyclonic shear is imparted on the background atmospheric flow. Consequently, the region of most frequent CRWB has been found to occur to the north of the EDJ mean-state, in the Northwest Atlantic region (Weijenborg et al., 2012). These regions are similar to those for ACRWB and predominantly to the south of the EDJ mean-state. A hypothesis has been developed to explain the importance of CRWB in the Northeast Atlantic/ European region. It will be shown that the majority of Northeast Atlantic/ European CRWB events, which were found to significantly increase UK PM10], are dependent on the prior occurrence of ARCWB."*

**10. Page 12, line 10:** I think you mean [PM10] > $\overline{[PM10]}$ + 10 $\mu g \, m^{-3}$. However, please

check the major comments on this issue.

*This inequality has been corrected in the text.*

**11. Discussion**: Would be good for the author to extend the outlook in their discussion:

**11.1** From this work I could also conclude that this should be valid not only for PM10, but also for PM2.5, which are by far more influenced by transport due to their lower settling velocities. Do the authors have any opinion on that?

*This is a good suggestion and therefore a brief discussion has been included on P13L41*

*"In this study, PM10 advection from Europe is hypothesised to greatly influence UK [PM10] episodes. A potential extension for this study could be to analyse the relationship between the smaller PM2.5 and RWB. PM2.5 is a smaller and subsequently lighter particle than PM10, with a reduced gravitational settling velocity. Consequently, PM2.5 is more readily advected than PM10 and RWB may therefore be more influential in facilitating the advection of PM2.5."*

**11.2** Could the author extend the manuscript or discuss the absolute frequency of European CRWB?

*We feel that sufficient literature has been included, for readers to find this information; for frequency Fig. 2 in Masato et al., 2013 and for distribution Fig. 5 in Weijenborg et al., 2012.*

---

## Author Comment (AC2) · 4 Nov 2016

**Response to the minor comments made by Anonymous Referee 2**

**1. Abstract:** Please include the probability of PM10 exceedences both for days without RWB and for those conditions most likely to lead to an episode. Also make it clear that an exceedance when there is RWB is 3 times more likely than period without RWB (it is not clear currently what is 3 times more likely than).

*The probability of exceeding a hazardous UK [PM10] threshold has been included within the abstract. Furthermore the probability of exceeding a threshold for Omega Block events has been included. These steps will help to elucidate what is three times more likely that what.*

[Figure]

**2. Introduction 1):** It would be helpful to expand the description of what is meant by Rossby Wave breaking here. Start by a quick reminder of what a Rossby Wave is and then give a bit more explanation of what is meant by large scale overturning (and that it is not overturning in the vertical!). A figure similar to Figure 2 of Masato et al., 2012 would be useful to better orient the reader and also help explain the diagnostics in section 2.

*The following text has been inserted at the beginning of the paragraph beginning on P2 L33:*

*"Synoptic-scale baroclinic eddies lead to wave-like distortions of the subtropical jet and to wave-breaking regions on the poleward and equatorward sides of the jet (known as RWB) (Haynes, 2015)."*

*The following reference was included:*

*Haynes, P. H.: CRITICAL LAYERS, In Encyclopedia of Atmospheric Sciences, edited by James R. Holton, Academic Press, Oxford, 582-589, doi: http://dx.doi.org/10.1016/B0-12-227090-8/00126-3, 2003.*

*We feel that adding a physical representation of what RWB is, would provide the reader with greater insight into RWB and overturning than a graphical representation of the metrics used to diagnose RWB.*

*P2 L33 has been changed to read:*

*"RWB is the large scale meridional overturning of air masses in the upper troposphere"*

*P2 L47 a line has been added to explain that RWB is diagnosed as the meridional overturning of potential temperature on the 2-PVU surface.*

**3. Introduction 2):** When discussing the way that high pressure influences concentrations, is not the suppression of vertical mixing by large scale subsidence also a factor which may play a role?

*We agree that this is an important part of how high pressure influences pollutant concentrations within the boundary layer. The following was included to explain this mechanism, on P2 L35:*

*"High pressure is directly associated with the elevation of PM10 concentration, through the suppression of vertical mixing out of the boundary layer."*

*The subsequent text on P2 L35 "This high pressure anomaly can influence" has been changed to read "This high pressure anomaly can also influence".*

**4. Section 2.1:** You state ERA-Interim data has been temporally filtered, but has it also had a running mean in longitude applied, as described in M11 and M13? If so please state this, if not explain why.

*This is the case and has been included in the final text. P4 L31:*

*"As in M13 a $15^o$ longitudinal running-mean filter has been applied to the calculated fields for $\overline{\theta}_{i,j,t}^{n}$ and $\overline{\theta}_{i,j,t}^{s}$. The longitudinal filter removes the influence of small-scale transient features on the calculation of the DB and RI indices."*

**5. Section 2.2 1):** I found the section explaining the exceedance threshold confusing. Please remove the first sentence and start the paragraph with "In this study PM10 exceedances are defined using a threshold based on the results of the European Study of Cohorts for Air Pollution effects (ESCAPE)..."

*This revision has been made, while a further alteration has been made to motivate the hazardous UK [PM10] threshold using the APHEA2 project, as opposed to the ESCAPE project. This alteration was made following a comment made by anonymous reviewer 1 (See review 2).*

*This paragraph now begins:*

*"In this study PM10 exceedances are defined using a threshold based on the results of the APHEA2 project."*

**6. Section 2.2 2):** At the end of the paragraph replace the last part of the final sentence (after = 2.98) with: Therefore we use a threshold for daily mean [PM10] of 29.72 $\mu g$ $m^{-3}$ or $\log_e$[PM10]=3.39 to define an exceedance.

*In the sentence beginning: "The tri-site", the dependent clause following "2.98," has been removed. The suggested sentence has been added, with the minor alteration that $\log_e$[PM10] is replaced by ln[PM10]. This alteration is made following the suggestion from anonymous reviewer 1 (See review 7) and has been made throughout the entire text.*

**7. Section 2.2 3):** It would also be useful to put this threshold in context by comparing to EU air quality standards and the UK DAQI for example.

*The EU legal threshold, which is also applicable in the UK has been mentioned on P5 L61. The text reads:*

*"For comparison, the European legal daily mean UK [PM10] threshold is currently set at 50 $\mu g$ $m^{-3}$ and must not be exceeded more than 35 times a year (European Union, 2008)."*

*One reference was subsequently added:*

*Council of the European Union and Parliament of the European Union (2008). Directive 2008/50/EC of the European Parliament and of the Council.*

**8. Section 3.3:** The sentence starting: "Fig 4 illustrates" that lead to UK PM10 exceedance the following day is unclear. Is the PM10 exceedance one day after the MSLP anomaly (2 days after the RWB event) or one day after the RWB event? Please clarify. If it is the former, please give more information on why a lag between PMSL and PM10 was used.

*The 1 day lag that exists between RWB and MSLP, also exists between RWB and UK [PM10]. The sentence has been altered to clarify this important point.*

*"the following day" has been replaced by "the day following RWB".*

**9. Section 3.4 1):** Please include a new paragraph after paragraph 2 to introduce the CDFs here. Describe Figure 5, focussing first on the blue and black lines. Then move on to the importance of persistence. It might even be useful to have 2 separate subsections for these.

*A paragraph was added to introduce the concept of CDFs and specifically what Fig. 5 illustrates, P8 L81:*

*"To illustrate the probability of exceeding a UK Midlands [PM10] threshold, Fig. 5 illustrates four cumulative distribution function (CDF) plots. The CDFs in Fig. 5 present the probability of exceeding any ln[PM10] value, for three subset [PM10] datasets. The first dataset (blue in Fig. 5), relates to days where no RWB of any type was detected within the region of RWB influence for that RWB subset. The black line in Fig. 5 represents days where RWB of the subset being analysed has occurred, following a day of no RWB (defined as onset RWB events). The red line represents continuous RWB events where RWB of the subset being analysed has followed a day of RWB of any type."*

*Text from "Northeast Atlantic/ European RWB..." on P8 L85 to "within a region of influence (red)" on P9 L91 has been removed.*

**10. Section 3.4 2):** This analysis only covers events with persistence of 1 day. Have longer periods of persistence been considered or are there too few of these for statistical significance.

*Due to the spatial constraint placed upon RWB in this study, events with a persistence of longer than 1 day, for instance 5 days (as in Masato et al., 2013), are too infrequent to generate statistically significant results.*

**11. Section 5.2:** Would it be appropriate to present the figures from the other observations sites as supplementary material.

*Supplementary material has been added, which looks at the correlation between*

*[PM10] at the three UK Midlands air quality monitoring sites. The tri-site super site [PM10] is representative of all sites. Therefore an extension of the work undertaken for all of the subsites, we feel, is not included.*

**12. Technical Corrections**

*The below listed corrections have been made to the text*

**12.1 P5, L59** *followed -> follows*

**12.2 P7, L49** *Subsequently -> This ensures that*

**12.3 P8, L71** *prevalent -> favourable*

**12.4 P9, L11** *Subsequently -> Therefore*

**12.5 P9, L11** *Is pre-determined from -> depends upon*

**12.6 Missing Reference:** *Buchholz et al., 2010 has been inserted within the references list*

---

## Author Comment (AC3) · 4 Nov 2016

**Response to the minor comments made by Anonymous Referee 3**

**1. P2, L54-55:** "The pressure dipole results from the meridional advection of upper level air masses with anomalous potential vorticity (PV) characteristics." I think that the link between upper-level PV and the pressure dipole is not immediately clear. A clarifying sentence might help.

*A clarification section has been added prior to this sentence, it reads:*

*"A region characterised by anomalously low PV, i.e. cyclonic motion, is associated with upper tropospheric divergence and subsequent convergence in the lower troposphere. Such a pattern is associated with anomalously low MSLP at the surface. The opposite*

[Figure]

*signal can be seen for anomalously high PV or anticyclonic motion."*

**2. Section 2.1:** Please move the formulas to the place in the text where they are referred to. At the moment, for instance, formula (6) is appended at the end of the section, but discussed at the top of page 4.

*Formulae (4) and (5) have been moved, to follow the text:*

*"resulting in CRWB/ACRWB occurring respectively".*

*We feel that formula (6) is presented most clearly in its current positioning.*

**3. P4, L41:** At the three measurement sites, being classified by DEFRA as urban background sites, are 'justified' (motivated) by the fact that the majority of the UK's population live in urban areas. Further down (L44-48) it is discussed that the sites are influenced by 'urban' activities, and that, therefore the three stations are combined to remove local spikes. How does this fit together with the motivation? By the way: What is DEFRA?

*P4 L41 has been altered to:*

*"The three sites used are all classified by the Department of Environment, Farming and Rural Affairs (DEFRA) as urban background sites, each representative of their urban environment, away from direct anthropogenic sources (DEFRA, 2014). "*

*It has been clarified on P4 L44 how the spikes removed are related to the influence of direct PM10 point sources on each urban background air quality monitoring station, which are placed to avoid direct influence from individual point sources (DEFRA, 2014).*

**4. P5, L53 and L61:** "A daily mean [PM10] ([PM10]) exceedance has been defined in this study, when [PM10] exceeds the threshold of 29.72 $\mu$g m$^{-3}$ (log$_e$[PM10] = 3.39)" "The tri-site [PM10] is 19.72 $\mu$g m$^{-3}$ or log$_e$[PM10] = 2.98, resulting in an impact threshold of 29.72 $\mu$g m$^{-3}$ or log$_e$[PM10] = 3.39": Repetitive?! Further, why does this, in the

second sentence, result in an impact value?

*In the sentence beginning: "The tri-site" the dependent clause following "2.98," has been removed. The following sentence has been added:*

*"Therefore we use a threshold for daily mean [PM10] of 29.72 $\mu g\ m^{-3}$ or ln[PM10]=3.39 to define an exceedance. "*

*Furthermore $\log_e$[PM10] is replaced by ln[PM10] throughout the whole text, following a suggestion from anonymous reviewer 1 (See review 7).*

**5. P5, L62-70:** Here the time-lag issue is discussed. It is argued that a time lag between [PM10] and RWB makes sense. But, no time lag is used for negative BI and [PM10]. Finally, the whole paragraph starts with the finding that the best correlations result if no time lag is used between RWB and MSLP. The reader can easily get lost in these many different cases! A little remedy could be if the link between MSLP and RWB is not discussed. To me it sounds rather obvious that the best correlation occurs if no time lag is used. And, by the way, I don't see a need to motivate the RWB time lag by a corresponding MSLP time lag. In short: Simplify the paragraph a little!

*Discussion of the temporal lag between RWB and MSLP has been removed. What remains is an explanation for the 1-day lag between positive BI values and UK [PM10] and the 0-day lag between negative BI values and UK [PM10].*

*The paragraph now reads:*

*"The concept of a temporal lag between RWB and UK [PM10] is explored during the analysis. The greatest UK [PM10] is found when using a one-day lag following BI>0 K km $^{-1}$. The lag accounted for the time taken for European PM10 to advect into the UK and for the UK to subsequently be exposed to a new air mass. In events where RWB was not diagnosed (BI<0 K km$^{-1}$), it was found that a 0-day lag provided the best relationships between RWB and [PM10]. Negative BI values are associated with westerlies entering the UK, providing the most efiňĄcient [PM10] removal processes.*

*These removal processes reduce surface [PM10] on timescales of less than a day and hence there is a lag of 0 days between negative BI values and resultant UK [PM10]. Therefore in Sect. 3.1 and 3.2, when a day is not indicating RWB (BI>0 K km$^{-1}$), no temporal lag was applied."*

**5.1:** To me it sounds rather obvious that the best correlation occurs if no time lag is used

*In the case of high pressure it is seen that the greatest UK [PM10] is found using a one-day lag following BI>0 K km$^{-1}$. This temporal lag allows for the time taken to transport PM10 from mainland Europe into the UK.*

**6. Section 3:** This section contains the main results from the study. All in all, the discussion is clear and the results are well supported by the data. However, while reading from subsection 3.1 to 3.2, to 3.3 and finally 3.5 I got a little lost. Many aspects of the link between [PM10] and RWB, positive and negative BI, CRWB and ACRWB, the exceedances of [PM10]... are discussed. I think it would be great to start the whole section 3 with a (rather short) introductory paragraph that, from the beginning, tells the reader here the journey will go to. In short: Give the reader some guidance what he can expect from this section and how the different subsections are connected.

*An introductory paragraph has been added to Sect. 3, which will act to aid the reader through the results section of this study.*

*"Section 3 presents the main results from this study. Section 3.1 begins by analysing the relationship between BI and UK Midlands [PM10]. Section 3.2 presents regions where RWB occurrences result in significantly elevated UK [PM10]. This analysis is undertaken for four RWB subsets; Warm Anticyclonic, Cold Anticyclonic, Warm Cyclonic, and Cold Cyclonic RWB. Section 3.3 analyses the MSLP response to both ACRWB and CRWB. The most important result from this study is presented in Sect. 3.4 and refers to the probability that days on which RWB occurs, lead to hazardous PM10 threshold exceedances. A RWB subset that results in the greatest probability of exceeding a*

*hazardous UK [PM10] threshold is examined in more detail in Sect. 3.5. The mechanism dictating the occurrence of Northeast Atlantic/ European CRWB is presented in Fig. 6 and discussed from Sect. 3.5 onwards."*

**7. P6, L20:** move formula (7) and (8) to the place in the text where they are referred to.

*We feel like the current placement of formula (7) and (8) present the clearest possible presentation.*

**8. Figure 2 and corresponding text:** In panel b) and c) there are some data points at rather low BI values. I wonder whether there data points, with a considerable leverage, influence the overall fit of the least-square fit? How does the correlation change if these points are omitted? I am also not perfectly convinced that it is reasonable to look at BI> 0 points only (red points and curve fitting) and to deduce that, for instance in panel b), the BI>0 has no impact on [PM10]. Finally, it might be better to use for all four panels the same range for the x axes. This would allow the different locations (GP1-4) to more easily be compared.

*A sensitivity analysis was undertaken, to analyse the impact of the outliers. Any BI values < 30 K km$^{-1}$ were masked out of this analysis and the correlation coefficients obtained from this analysis. This sensitivity analysis has been described within the text on P6L96:*

*"Fig. 2 b) and c) are influenced by very low BI values, which appear as outliers in the general distribution. It was shown that by removing BI values < -30 K km$^{-1}$, there was very little impact on the overall CC values obtained from Fig, 2 b) and c). In Fig. 2 b) and c) the Correlation coefficient increases from 0.42 to 0.43 and from 0.32 to 0.33 respectively, following the removal of the outliers."*

*The red points in all subplots in Fig. 2 provide a data subset that is independent from the blue dots. Therefore it is reasonable to analyse this dataset separately from the blue dots. If this step is taken a correlation coefficient of 0.04 exists between BI and*

[Figure]

UK [PM10] for BI>0 K km$^{-1}$ in Fig. 2b). *Therefore in this analysis, the magnitude of BI>0 K km$^{-1}$ has very little, if any impact on UK [PM10].*

*The x-axis scales in Fig. 2 have been altered as to be consistent throughout all four subplots.*

**9. P7, L24-25:** "the following longitudinal filter has been applied: 277.5$^o$E < longitude < 77$^o$E in order to focus on regions influential upon UK [PM10]." I do not clearly understand what you mean with longitudinal filter? Do you simply neglect all RWB events in this domain?

*The sentence has been re-phrased to read:*

*"To focus on the regions that are shown to be influential to UK [PM10], a longitudinal mask has been applied so that only longitudes east of 277.5$^o$E and west of 77$^o$E are included in this analysis."*

*For clarification, only RWB events that have occurred within the selected domain have been analysed.*

**10. P7, L30-31:** Incomplete sentence

*This has been addressed to read:*

*"Figure 3 shows that warm and cold CRWB significantly raise UK [PM10], predominantly in the Northeast Atlantic/ European region and not in the Northwest Atlantic region."*

**11. P10, L33-35:** Repetitive?! Is it the same 10-grid point criterion used before? If so, I would prefer this criterion is introduced only once in the text.

*The sentence re-iterating the 10 grid point threshold has been removed. P10 L33-35*

**12. P12, L10:** "[PM10]>[PM10] + 10 $\mu$g m$^{-3}$" Unclear!

*The overbar has been added to the first term on the left hand side. This now reads:*

"[PM10]> $\overline{[PM10]}$ + 10 $\mu$g m$^{-3}$"

**13. P12, L15:** "The greatest probability of exceeding this studies hazardous [PM10] threshold was associated with a synoptic mechanism identified by an Omega Block (probability 0.383)" Complicated sentence! Please rephrase.

*The sentence was rephrased to read: "In this study, the Omega Block mechanism resulted in the greatest probability of exceeding a hazardous [PM10] threshold."*
* * *